# Characterising Carbon Monoxide Household Exposure and Health Impacts in High- and Middle-Income Countries—A Rapid Literature Review, 2010–2024

**DOI:** 10.3390/ijerph22010110

**Published:** 2025-01-15

**Authors:** Sarah V. Williams, Rebecca Close, Frédéric B. Piel, Benjamin Barratt, Helen Crabbe

**Affiliations:** 1UK Field Epidemiology Training Programme, UK Health Security Agency (UKHSA), London E14 4PU, UK; 2Environmental Epidemiology Team, Radiation, Chemical and Environmental Hazards Directorate, UK Health Security Agency (UKHSA), Didcot OX11 0RQ, UK; 3National Institute for Health and Care Research Health Protection Research Unit (NIHR HPRU) in Environmental Exposures and Health, Imperial College London, London W12 OBZ, UK; f.piel@imperial.ac.uk (F.B.P.);; 4Small Area Health Statistics Unit, MRC Centre for Environment and Health, Department of Epidemiology & Biostatistics, School of Public Health, Imperial College London, London W12 OBZ, UK; 5Environmental Research Group, MRC Centre for Environment and Health, School of Public Health, Imperial College London, London W12 OBZ, UK

**Keywords:** carbon monoxide, household exposure, exposure models, literature review, indoor environments, public health

## Abstract

Carbon monoxide (CO) is a toxic gas, and faulty gas appliances or solid fuel burning with incomplete combustion are possible CO sources in households. Evaluating household CO exposure models and measurement studies is key to understanding where CO exposures may result in adverse health outcomes. This assists the assessment of the burden of disease in high- and middle-income countries and informs public health interventions in higher-risk environments. We conducted a literature review to identify themes that characterise CO exposure in household dwellings. A keyword-structured search using literature databases was conducted to find studies published in the period of 1 January 2010–5 June 2024. We focused on studies from high- and middle-income countries, excluding animal and biomass studies, and narratively synthesised themes. We identified 5294 papers in the literature search and included 22 papers from thirteen countries in the review. Most measured CO levels were below the WHO or country guidance levels, with sporadic peaks of measured CO linked to fuel-burning activities. To understand CO exposure in households, we identified sixteen themes grouped into five main categories: dwelling characteristics, source characteristics, temporal variation, environmental characteristics, and socioeconomic status of occupants. Seasonal variation (temporal variation), size of room and ventilation (dwelling characteristics), and cooking and outdoor CO levels (source characteristics) had the most evidence. These themes characterising CO exposure in household dwellings are important to aid the development of indoor exposure models and for understanding where CO exposures result in adverse health outcomes. These themes should be validated by household CO monitoring studies, which will enable the identification of higher-risk household dwellings and inform public health actions.

## 1. Introduction

People spend an estimated 80–90% of their time indoors [1] with indoor air pollutants representing a neglected health risk [2]. Carbon monoxide (CO) is a pollutant present in outdoor and indoor environments, formed as a by-product from the incomplete combustion of fossil fuels or biomass sources. Depending on the severity and duration of exposure, symptoms can include headaches, flu-like symptoms, confusion, permanent neurological damage, or death [3]. Non-specific symptoms and differential diagnoses make it difficult to diagnose CO poisoning and, hence, characterise the true burden of disease from CO exposure.

In 2021, an estimated 28,900 people globally died from accidental or unintentional non-fire related (UNFR) CO poisoning, with an estimated 536 deaths in Western Europe [4]. Globally, nearly 70% of deaths occurred in males, with the highest mortality rate in those aged 85 and older and the highest age-standardised mortality rate occurring in Eastern Europe [4]. Whilst there have been improvements in CO mortality rates in recent years, these are inconsistent across regions [4]. According to the UK’s Cross Government Group On Gas Safety And Carbon Monoxide (CO) Awareness 2021–2022 annual report, there were approximately 20 UNFR CO deaths per year between 2017 and 2021 in England and Wales [5]. Most of these deaths are preventable. A review of narrative coroner verdicts of 750 UNFR CO deaths in England and Wales between 1998 and 2019 gave contextual details of the circumstances around these deaths. The authors found that 77% of deaths occurred in males, with 59% occurring in a home dwelling, and 68% of deaths occurring in autumn or winter, which was likely due to the risk of CO poisoning from heating sources without adequate ventilation [6].

Whilst global data on hospitalisation due to UNFR CO poisoning is patchy, in the USA between 2005 and 2018, there were an estimated 19,230 hospitalisations, a rate of 7.3 per 1,000,000 [7]. In Canada, there were 1985 hospitalisations in 1995–2010, a rate of 3.85 per 1,000,000, with admissions most frequent from September to April [8]. The average UNFR annual hospital admission rate for CO poisoning between 2002 and 2016 in England has been estimated as 4.1 per 1,000,000 [9].

For both hospitalisations and deaths, these numbers are likely to be underestimated due to misdiagnosis, a lack of testing, cases not accessing healthcare, and the short half-life of in vivo CO, making it difficult to test for even if CO poisoning is suspected [10]. In the UK, ambient household levels of CO are reported to be low and below the World Health Organization’s (WHO) recommended health-based guidelines [11,12]. However, there is still a risk of chronic low-level exposure in the home, which may be associated with respiratory and cardiovascular morbidity [12]. Hence, CO poisoning is likely to have a bigger public health impact than the official figures show.

To understand the burden of disease from exposure to indoor air pollutants, it is important to understand the level of contaminants and the implications of exposure. A literature review of indoor air pollutant concentration levels, including CO, by Karakitsios and colleagues identified seven studies that identified key risk factors for CO exposure [13]. In most homes in the studies, CO concentration levels were below 1 ppm (1.165 mg/m^3^) over the monitoring period [13], which is below the WHO’s recommended health-based guidelines of 4.4 ppm (5.1 mg/m^3^) over 24 h [11]. Higher CO concentration levels were seen in homes with open fires and gas cookers and among residents who smoked. A UK study included in the review focused on the effects of different types of residential heating where CO levels remained below the detection limit of 1 ppm (1.165 mg/m^3^) [14] for all types of heating [13,15]. A Greek study demonstrated that good ventilation lowered high indoor CO concentration levels [13,16]. Seasonality of CO exposure has also been well demonstrated, with poisonings more common in the heating months when ventilation is reduced and fossil fuel burning appliances are in use [7,9,17].

Therefore, identifying themes that characterise CO exposure in household dwellings is key to understanding where CO exposures may result in adverse health outcomes, including chronic low-level exposure. Exploring these themes can help to inform public health inventions in higher-risk environments. Much of the housing in low-income settings is likely to be different to those in high-income settings, such as the United Kingdom, as fuel sources for heat and cooking vary. Therefore, we wanted to identify studies that had characterised household CO exposure relevant for high- and middle-income settings.

We conducted a rapid literature review with the following objectives:(i)To identify studies that have measured or modelled CO concentration levels in households and the factors that have affected those levels;(ii)To identify and summarise themes characterising higher CO levels in terms of human health exposure in household dwellings in high- and middle-income countries (HMICs);(iii)To describe models developed or used to understand CO household exposure, with the intention of using these to aid interpretation in epidemiological studies of CO household exposure;(iv)To summarise studies that estimate the health impacts of CO household exposure and the factors used to characterise the health consequences of exposure, such as in use for health impact assessment (HIA).

This article is a revised and expanded version of an abstract entitled “Characterising carbon monoxide exposure in household dwellings in middle- and high-income countries—a literature review”, which was presented at the 36th Annual International Society for Environmental Epidemiology Conference, Santiago Chile, 25–28 August 2024 [18].

## 2. Methods

### 2.1. Identification of Studies

Keyword-structured searches were performed using Ovid MEDLINE, Ovid Embase, Web of Science and Google Scholar for peer-reviewed articles published between 1 January 2010 and 5 June 2024. These dates were chosen to ensure that the selected literature was most representative of households and heating and cooking habits of current times. The search was conducted in March 2023 and repeated in June 2024 to bring it up to date. See Appendix A for the search strategies for the different databases.

### 2.2. Inclusion Criteria

We included studies in this review that met the following criteria:Contained modelling or monitoring of CO concentration levels, sources, emissions, or related health effects;Attempted to build, use, validate, or evaluate a model for such exposure or for its use for human health exposure assessment and health impact assessment;Were set in the indoor home environment;Were set in a high or middle-income country (HMIC).

We had no language restrictions. Our preference was for studies that have measured or modelled CO in multiple homes. However, studies in single homes were included if they otherwise met the inclusion criteria.

#### Exclusion Criteria

We excluded animal studies and studies that only focused on outdoor levels of CO, symptoms of CO poisoning, personal exposure rather than household exposure, characteristics of CO poisoning cases, or focused on knowledge, attitudes, or beliefs about CO poisoning. Studies from low-income countries [19], those focused on informal settlements, or those that focused on biomass burning were excluded, as these were thought to be less likely to reflect the characteristics of CO exposure in high- and middle-income countries. We excluded the following sources of information: conference abstracts, consensus statements, guidelines, study protocols, commentaries, preprints, and letter responses. Studies were also excluded if they were written in a language that our translation services could not translate.

### 2.3. Selection of Studies

A pilot review of a portion of returned papers, 20%, was conducted based on title and abstract by two authors (SVW, RC, and HC) in total, equally dividing the papers between them. Any disagreements on inclusion and exclusion were discussed by the three reviewers (SVW, RC, and HC), with a majority decision being the final arbiter on inclusion. The lead author (SVW) reviewed the remaining papers based on title and abstract and then on full text. Any papers that they were uncertain about whether to include based on title and abstract were discussed with the review team (RC and HC) and for full text with one other co-author. For title and abstract, a majority decision was the final arbiter on inclusion, whilst consensus was reached for full-text papers.

### 2.4. Assessment of Study Quality and Data Extraction

Data were extracted by the lead author (SVW) from each included study using the Excel template in Appendix A. Assessment of study quality was performed using the MetaQAT tool [20]. This critical appraisal tool has been developed specifically for public health studies and can account for a wide diversity of study designs.

### 2.5. Synthesis of Data

Due to the heterogeneity of the studies, it was not possible to pool estimates and conduct a formal meta-analysis. Study data were narratively synthesised into themes by the lead author, with common themes identified; the results and information from the studies were extracted and synthesised into a summary of each theme. The themes were grouped into categories. The characteristics of the studies, such as country and setting, were extracted, as were measurements of CO concentration levels.

### 2.6. Patient and Public Involvement

There was no patient or public involvement in the design of this study.

### 2.7. Ethical Approval

No ethical approval was required. This review was based on published data and did not involve human subjects or patient-identifiable information.

### 2.8. Protocol

This rapid review was carried out according to the PRISMA (Preferred Reporting Items for Systematic Reviews and Meta-Analyses) framework [21]; see Appendix A for the completed PRISMA checklist.

## 3. Results

The search returned 5294 papers, and 3332 remained after the removal of duplicates. Following the review stages (see Figure 1), 22 papers were included in the rapid review.

### 3.1. Characteristics of Studies

Out of the 22 publications included in the review, most studies (77%) were observational in design and focused on urban settings (55%) and came from thirteen different countries, with 32% of the studies from the USA (see Table 1 and Figure 2). The studies were set in different types of housing and focused on different rooms in the dwelling. An overview of each study, including outcomes and themes identified that characterise CO levels, is included in Table 2. Figure 2 shows the countries included in the review and countries where relevant studies were found.

### 3.2. Range of Measured CO Concentration Levels in Households

The CO concentration levels measured in households in the studies included are summarised in Table 2. The frequency and duration of measurements varied between studies, with some studies measuring CO concentrations over a few minutes whilst others measured over several days. Most measurements were below the WHO or country guidance levels (see Table 3 for WHO guidance levels), although spikes in CO concentration levels were seen in some studies related to cooking activities (12–22 ppm [38] and 20–30 ppm [36]) and during coal combustion (mean values of 85.4 ± 80.6 ppm [35]).

### 3.3. Categories and Themes Identified from the Studies

The five categories and sixteen themes identified from the studies are summarised in Table 4.

#### 3.3.1. Temporal Variations


*Seasonal variations*


There was some evidence demonstrating seasonal variations in CO concentration levels with higher levels of CO in the winter months [28,37,39], including a study of thirty homes in rural Romania and Slovakia [28] and five apartments in London [37]. In the study in London, the seasonal variations in indoor CO concentration levels were very small, with overlapping confidence intervals [37]. Larger variations were seen in the outdoor CO concentration levels, with higher levels seen in the heating season (January–February) compared to the non-heating season (June–September) [37].

A study of 400 homes in Tehran, Iran, used outdoor air quality data, daily pattern of natural gas burner usage in homes and the ventilation rate in residential buildings to model CO concentration levels [39]. Higher concentration levels of CO were demonstrated in the winter and autumn months than in spring and summer, reflecting different seasonal heating usage and habits [39]. This study measured ventilation rates by grouping the homes into different categories and releasing CO tracer gas in three locations for each category in both summer and winter. The mean ventilation values in summer and winter were 1.55 ± 0.27 and 1.22 ± 0.22 Air Changes Per Hour (ACH) [39]. The authors hypothesised that the higher ventilation rate in summer could be due to more natural ventilation and air conditioning (AC) than heating systems in winter [39], although the confidence intervals of the mean seasonal measurements overlapped. The modelled mean CO indoor concentration levels were higher in autumn than spring, and whilst the ventilation rates did influence this, the authors report that the biggest effect was from outdoor CO concentration levels [39]. However, an Australian study of 111 homes found no relationship between airing homes in either winter or summer (the opening of doors and windows for fresh air) and CO concentration levels in suburban non-smoking homes [27]. In this study, CO was measured once over a continuous 24 h period with monitors placed in the living rooms of homes, likely far away from home CO-producing sources; CO was not monitored outside, and mean indoor 24 h CO household concentration levels were low (0.94 ± 0.1 ppm) [27]. Winter and summer measurements were not repeated in the same home. Hence, it is possible that fluctuations in CO concentration levels due to ventilation and season were not detected in this study.


*Daily variations*


There were relatively few studies that considered daily variations in detail, as many did not take CO measurements throughout the day. However, a study in Iran observed that the highest average of CO concentration levels in homes was modelled to be between 21.00 and 22.00 h, corresponding with the second highest frequency of natural gas burner use from participants’ questionnaire responses [39]. In a UK study of three homes, CO concentrations were measured in the kitchens, with peaks related to cooking times [38]. Other studies measured CO continuously across a 24 h period but did not report fluctuations in CO concentration levels and only reported mean values over a 24 h period [27].

#### 3.3.2. Environmental Characteristics


*Temperature*


There was limited evidence on the relationship between temperature and CO concentration levels. A statistically significant association was reported between mean CO levels in the living room and temperature, both ambient and indoor, in a study in Egypt [22]. The authors hypothesised that this could be due to the thermal gradient between outdoor and indoor environments forcing warm air inside [22]. Another study in the USA examined the relationship between CO concentration levels and temperature, but no correlation was identified [34]. Two studies measured temperature and CO concentration levels, but the relationship was not explored further [25,28].


*Relative humidity*


There was limited evidence on the relationship between humidity and CO concentration levels. A statistically significant relationship was demonstrated between indoor and outdoor relative humidity and CO concentration levels in the living rooms of homes in one of the two study areas, where most of the homes were naturally ventilated [22]. No relationship was observed between CO concentration levels and indoor or outdoor relative humidity in the other study areas where most homes were mechanically ventilated [22]. Another study in the USA examined the relationship between CO concentration levels and relative humidity, but no correlation was identified [34]. One other study measured relative humidity and CO concentration levels, but the relationship was not examined [25].

#### 3.3.3. Dwelling Characteristics


*Floor level*


There was a lack of studies examining the relationship between CO concentration and the floor level of the home. A study in Portugal of 552 homes characterised indoor air quality, including CO, and collected information on the floor levels of homes [25]. However, the relationship between CO concentration levels and the floor level of the home was not examined further.


*Size of rooms*


There was some evidence exploring the relationship between the size of a room and CO concentration levels, with CO concentration levels higher in smaller rooms. A study in Egypt of 60 flats demonstrated a significant negative correlation between kitchen size and mean CO, with CO accumulating in smaller kitchens, which were classed as those under 20 m^3^ size [22]. CO concentration levels in the kitchens were significantly correlated with levels in living rooms, and a significant negative correlation was also demonstrated between the size of living rooms and CO concentration levels [22]. Whilst another study collected data on the mean size of homes, the relationship with CO concentration levels was not explored [26].

In a Chinese study in one naturally ventilated home, the CO concentration levels in all the rooms were correlated, with the highest concentration measured in the kitchen [36]. There was a lag between the rooms, related to the time required for the CO to disperse from the emission source, which was associated with distance from the kitchen (with a negative dependence of association on distances longer than 5.5 m) [36]. CO concentrations also varied with vertical height, which the authors thought was due to rising hot air from cooking [36].


*Number of occupants*


There were two studies that explored the relationship between the number of household occupants and CO concentration levels. A statistically significant correlation was demonstrated between the number of occupants and CO concentration levels in a study of 60 flats in Egypt [22], which the authors hypothesised was due to increased human activities. However, in an Australian study of 111 homes, no relationship was identified between CO concentration levels and the number of occupants [27].

There was a lack of supporting evidence from other studies which, whilst they collected data on household occupancy, did not examine the relationship with CO concentration levels [25,28,31].


*Age of building*


One study of homes in Egypt identified a significant correlation between the age of the building and CO levels [22]. A more positive relationship was seen in the area where the mean age of buildings was older, with the authors hypothesising that in older buildings, outdoor pollution could infiltrate the house via cracks and crevices in the building [22]. Older homes might also be more likely to have older gas or fossil fuel-burning appliances. However, in countries like the UK, outdoor CO concentration levels are likely to be well below air quality guidelines [11], so infiltration from outside is likely to be less of an issue, although CO may diffuse from homes to outside, resulting in lower CO levels inside.

However, in an Australian study, no relationship was identified with the age of homes, categorised as either under 10 years old or over 10 years [27]. Whilst other studies collected data on household age, the relationship with CO concentration levels in the homes was not explored [25,26].


*Ventilation*


Ventilation affects the inflow of pollutants from outside and the removal of pollutants from indoors [44]. It is sometimes referred to as the air exchange rate (AER) in the literature, the airflow rate per volume of the indoor environment, or, more commonly, the air change per hour (ACH), which we use here. Some studies calculated the ACH in households by using a tracer gas (e.g., CO_2_, SF_6_) [23], and measuring its decay rate to understand the differences in CO concentration levels within households, between households, and in the outdoor environment.

Of the studies included in this review, 10 (50%) considered ventilation and its effect on CO concentration levels to varying degrees. ACH and mean CO concentration levels were found to be higher in homes that were naturally ventilated compared to homes with air conditioning (AC) [22], and ACH was affected by the season [39]. In a study in Egypt, 93% of homes in one of the two study areas were naturally ventilated, and the homes in this area had higher mean ACHs (3.3 ± 0.7 h^−1^) and mean indoor CO concentration levels than the other study area, in which 87% of homes used AC (ACH 1.2 ± 0.4 h^−1^) [22]. The stronger indoor–outdoor correlation for CO concentration levels in the study area with most homes naturally ventilated likely reflects the more frequent opening of windows and doors [22]. In this study area, the mean indoor-outdoor (I/O) ratios were also higher in both the living rooms and kitchens compared to the other study area, which the authors attributed to the higher ACHs and to more polluted air entering homes via open doors and windows [22].

Higher ACHs were seen with open doors and windows in a study focusing on ventilation patterns and indoor air quality in a single bedroom [23]. This study observed that an open door and window had the highest ACH (4.85 ± 0.57 h^−1^), and the lowest was with a closed door and window [23]. However, the lowest CO concentration level was when the door was open, and the window closed (mean 1.6 ± 0.3 mg/m^3^) (1.37 ± 0.26 ppm), and a closed window may stop CO infiltrating from outside [23]. In this study, the window was above a street of moderate traffic and a restaurant kitchen on the ground floor, possible outdoor CO sources, but in other settings with CO sources inside, a closed window could lead to higher concentration levels. The highest CO concentration level was when the door and window were closed (3.8 ± 1.0 mg/m^3^) (3.26 ± 0.86 ppm), although similar values were seen for the two other combinations of closed and open doors and windows [23]. A USA study in two homes also observed that the air change rate increased as windows were opened and, in one home, were higher for experiments during the day compared to the night, which the authors attributed to greater daytime wind speeds [42,43]. However, the relationship between air change rate and CO close to the source is complex [42]. A study on coal combustion in homes observed moderate air exchange (0.14 h^−1^) between indoors and an outdoor courtyard, removing some pollutants from inside [35]).

The use of air exhaust ventilation systems in kitchens increased the ACH [38,39] and reduced CO concentration compared with homes without air exhaust ventilation systems [22], and even occasional use of kitchen exhaust reduced CO peak concentration levels [31]. However, one study did not observe a relationship between CO concentration levels and using an extractor fan when cooking, although the CO monitor was in a different room [27], and in another study, the home with the largest ACH had the highest average CO concentration level [38].

Other studies collected data on the ventilation of homes, such as whether they were naturally ventilated or mechanical, but did not explore the relationship with CO concentration levels [25,29].


*Distance to roads*


There was mixed evidence on the relationship between proximity to roads and CO concentration levels in the homes. A study in Egypt demonstrated a negative correlation between CO concentration levels and proximity to main roads with heavy traffic, with a stronger relationship seen in one of the two study areas where most homes were naturally ventilated [22]. However, an Australian study that categorised homes as those that were within ≤300 m and those that were >300 m from a major highway identified no relationship with CO concentration levels in the homes [27], although this cut-off distance may have been too large to observe an impact on CO concentration levels in homes.

#### 3.3.4. Source Characteristics


*Cooking*


There were six studies that examined the relationship between CO concentration levels and cooking, with most studies observing higher CO concentration levels during cooking, particularly when natural gas was used as a fuel source.

Increases in the highest one-hour CO concentration levels in homes that used gas cooker burners were observed in a study in the USA, regardless of whether the gas appliances had ventilation [31]. Homes that used pilot burners also appeared to have increased highest one-hour CO concentration levels compared to those without [31]. However, when scaled for floor size, the impact of pilot burners was greatly reduced. Time spent cooking with gas appliances also increased the highest one-hour CO concentration levels, which was independent of floor area and was not observed for cooking with electric appliances [31]. Higher CO concentration levels were observed in kitchens where all homes used natural gas for cooking in a study in Egypt [22]. The CO concentration levels measured in the kitchens were strongly correlated with the levels measured in living rooms, demonstrating the contribution from cooking to indoor CO concentration levels throughout the home [22].

In a UK study of three homes, two homes using gas cookers observed spikes in CO concentration levels during cooking periods, whilst no spikes were observed in the home using an electric cooker [38]. These homes had the same average CO emission rate of 0.1 g.h^−1^, but the homes with gas cookers had CO emission rates of between 0.3 and 2.2 g.h^−1^ during cooking [38]. A study of one home in China also observed spikes in CO concentration levels during cooking, and natural gas was the primary fuel source [36]. Small spikes in CO were also observed with the use of electrical cookware [36]. Cooking emissions accounted for 49% of indoor CO compared with 51% of infiltration from outside based on floor area-weighted average [36].

In an Australian study, no relationship was identified between CO concentration levels and cooking, but the CO monitor was not placed in the kitchen [27]. Whilst studies in Chile and India collected data on cooking fuel sources, the relationship with CO concentration levels was not explored [29].


*Space heating and cooling homes*


There was some evidence of space heating and cooling homes from an Australian study of 111 homes. Most homes (41.4%) used reverse AC for heating, and the rest used gas, oil, electric, or a combination of methods. No relationship was identified with CO concentration levels [27]. For cooling homes, 9% of homes used fans, with the remainder using AC or a combination of AC and fans. Homes that used fans had higher concentration levels (*p* < 0.001) in a univariate analysis. In multiple linear regression, using only fans to cool the home was associated with significantly higher CO concentration levels (effect estimate: 0.221 ppm; 95% CI: 0.132, 0.310; *p* ≤ 0.001) [27].


*Fireplaces and stoves*


There was limited evidence on the relationship between the use of fireplaces and CO concentration levels, with five studies collecting data on fireplace use. CO concentration levels were greater than USA EPA and WHO guidelines (9 ppm 8 h average) in 20% of homes in a study of 30 homes with unvented gas fireplaces, which the authors note typically happened when the fireplaces were not installed to the manufacturer’s instructions [26]. The home that had the highest one-hour and eight-hour CO concentration levels had the fireplace in a ‘tight basement’(sic), and it was the sole heating source, which could lead to a build-up of CO concentration levels [26]. One study of a single wood-burning open fire observed that the mean concentration of CO was 0.1 ± 0.7 ppm during combustion, but during ash removal, the mean concentrations of CO increased to 40 ± 40 ppm with a maximum concentration of 132 ppm due to hidden embers [24]. In one study on coal burning in different types of stoves in rural China, CO concentration levels were significantly lower (*p* < 0.05) in the non-heating periods (2.8 ± 1.2 ppm) compared to the heating period (85.4 ± 80.6 ppm) and levels of CO increased during the combustion phase, peaking during the stable combustion stage [35]. Two studies measured CO concentration levels and collected data on fireplace and wood burning use, and nearly all CO measurements were below country guideline levels, but the relationship with CO concentration levels was not explored [25,28].


*Smoking*


There was a lack of studies exploring the relationship between smoking and CO concentration levels, although data was collected on smoking in three studies [25,28,29]. One study did observe that the mean indoor-outdoor ratios were higher in smoking than non-smoking homes, which was more marked in smoking homes in the lower socioeconomic status area of the two study areas [22].


*Outdoor air quality affecting indoor CO levels*


The evidence on the relationship between indoor and outdoor CO concentration levels was mixed, with some studies observing lower CO concentration levels outdoors than within homes and other studies observing higher outdoor CO concentration levels compared with indoors. Five studies in Iran, China, Chile, and the UK observed lower CO concentration levels outside. For example, in Chile, in 96 apartments, the outdoor CO concentration levels were observed to have diminished from 0.8 ppm in September to 0.44 ppm in December, and an inverse relationship was observed between outdoor and indoor CO levels (*p* = 0.106) [29]. The CO concentration levels outdoors and in kitchens in 25 homes in Iran were monitored for ten days each month between January and March; mean levels were higher indoors (0.84 ± 3.21 ppm) with outdoor CO concentration levels (0.27 ± 0.92 ppm) [32]. If the spikes in indoor CO concentration levels were removed from a study of one home in China, the baseline indoor CO concentration levels were associated with the outdoor CO concentration levels, but otherwise, levels indoors were higher [36]. A study of coal combustion in fireplaces in China observed that indoor concentrations were higher than outdoor courtyard concentrations and were significantly positively associated (*p* < 0.05) with outdoor levels and that the indoor combustion activities had the most significant contribution to courtyard pollution [35].

However, in two studies, the opposite was present. In a study in Egypt, mean 24 h outdoor levels of CO exceeded corresponding indoor measurements in the living rooms and kitchens of flats, with smaller differences between outdoor levels and levels in kitchens [22]. Correlations were seen between outdoor and indoor levels of CO, with weaker correlations for kitchens than living rooms, suggesting a higher contribution from indoor sources, such as cooking [22]. Stronger correlations were seen for homes in one of the two study areas, but more homes in this area were naturally ventilated compared to homes in the other area, which mostly had mechanical ventilation and air exhaust extraction in the kitchens [22]. In a different study, the CO concentration levels were higher outdoors than indoors in both winter and summer in five apartments in London [37]. The median outdoor CO concentration level across the study period was 294 µg/m^3^, and the mean was 330 µg/m^3^ [37]. The apartments had mechanical ventilation with heat recovery, and the air change rate was 0.5 h^−1^, with additional ventilation in summer provided by natural ventilation through opening windows [37].

#### 3.3.5. Socioeconomic Status of Occupants

There was little evidence of how the socioeconomic status of occupants affected CO concentration levels. Only two studies considered this aspect. In the Egyptian study, indoor levels of measured air pollutants, including CO, were higher in homes with lower income and education levels. However, this used a univariate analysis, and confounders were not considered [22]. A study of 316 homes in the USA collected data on household income, the highest education level of occupants and home ownership (54% of homes), but the relationship with CO concentration levels was not explored [31].

### 3.4. Models for Household CO Concentrations

There were seven studies that modelled CO, with three using a well-mixed mass balance model [40,42,43]. One of these studies modelled decay rates of CO near unvented gas fires [26] and another estimated the CO emission rate from cooking using CO concentration level measurements in kitchens [38]. A study of 400 homes in Iran estimated indoor CO concentration levels based on measured outdoor CO concentration levels, measured ventilation rates, and household questionnaire data on daily usage of natural gas burners combined with existing data on fuel consumption and emissions of natural gas burners [39]. The authors validated the model’s estimated CO concentration levels with measurements in one residential building hourly for 12 h with a linear regression coefficient of 0.8 [39].

The mass balance model was also used in a simulation study in the USA to estimate the population impact of cooking with natural gas burners on exposure to CO. The authors developed a population impact assessment modelling approach and used a representative sample of homes to estimate concentrations of CO [40]. Simulation models estimated parameters for each home in the sample, including environmental and energy performance factors and concentrations of CO in typical weeks of summer and winter [40]. These individual estimates were then compiled to develop population impacts, predicting that 4–9% of occupants in homes cooking at least once a week with natural gas burners and without the use of venting range hoods were exposed to CO concentration levels exceeding national (35 ppm 1 h average and 9 ppm 8 h average) and Californian guidelines (20 ppm 1 h average and 9 ppm 8 h average) [40].

The well-mixed mass balance model assumes that CO indoors instantly becomes well-mixed, which may be appropriate when the source emission and mixing time scales are much shorter than exposure duration and when not close to the source. A US study focused on understanding the proximity effect, i.e., the relationship between CO concentration level and distance from the source [42]. The higher CO concentration levels near the source are caused by the non-instantaneous mixing of the pollutant and surrounding air or microplumes [42]. Modelling exposure close to a continuous point source depends on three variables: (1) indoor concentration caused by outdoor air infiltration, (2) well-mixed concentration from the indoor source (mass balance model), and (3) contribution of the microplumes [42].

The same experimental work was analysed further to determine the turbulent diffusion air coefficient (K), which is how fast pollutants disperse from turbulent mixing indoors. There was reasonable agreement between modelled and measured averaged CO concentration levels, demonstrating that CO mixing in naturally ventilated rooms can be characterised and that exposures close to a source can be modelled [43].

Other models have been used in studies, such as a validated model developed by the American Gas Association Research Division, which “simulates how indoor air quality varies with time in a well-mixed space heated by a vent-free gas appliance” and estimates concentrations of pollutants using a well-mixed single chamber model [41]. This was used in a USA study that estimated the impact of vent-free gas heating appliances on indoor air quality, including CO, in USA homes constructed to be energy efficient [41]. Parameters including humidity, air exchange rate, appliance input rates, heat loss factors, and house and room volumes were varied to run simulations [41]. In all simulations, predicted CO concentration levels were below 9 ppm, which is the US Environmental Protection Agency’s standards [41].

### 3.5. Related Health Outcomes

Only one study considered health outcomes. CO and other airborne pollutants were measured in the kitchens and master bedrooms, and health questionnaires were completed in 552 homes across Portugal with the aim of assessing diagnoses and symptoms of rhinitis and asthma over the past year [25]. There was a positive association between asthma and symptoms of rhinitis and homes with higher levels of airborne pollutants (*p* < 0.01), but this was not specific to CO concentration levels [25].

## 4. Discussion

For our first two aims of identifying themes that characterise CO exposure in dwellings, six main categories with sixteen themes in total were identified from the literature with varying levels of evidence to characterise CO exposure in dwellings. The themes included dwelling characteristics, source characteristics, seasonal variation, daily variation, environmental characteristics, and socioeconomic status of occupants. The themes with the strongest evidence were those that had multiple studies that considered them; the individual studies were conducted in multiple dwellings, in which the relationship with CO concentration levels was explored, and the results of the different studies were consistent. This included seasonal variation, size of room and ventilation (dwelling characteristics), and cooking and outdoor CO levels (source characteristics), some of which were identified in the introduction as known risk factors for CO exposure. The other themes warrant being explored further to validate whether they have a relationship with CO concentration levels. Many of these studies measured CO concentrations, and reassuringly, most CO measurements were below WHO or country guidance levels, with sporadic spikes linked to household activities, such as cooking [22,36].

There was variation between studies about the theme conclusions, and some studies had findings opposite to what we would expect, such as distance to roads and daily variation in CO concentration levels. Some of these differences can be explained by the frequency of CO measurements, placement of CO sensors, whether measurements were repeated in the same households at different time points and outdoor CO concentration levels in the studies. Unfortunately, not all studies measured outdoor CO levels, which may have been raised in urban areas and could affect CO concentration levels measured in homes.

There are important themes that we would have expected to have been identified from the literature and which also warrant further investigation to fully characterise CO exposure in households. These include the effect of gas combustion boilers (water and heating systems) compared with other systems; differences in CO concentration levels between rooms and room use or purpose, since many studies did not measure levels in multiple rooms; infiltration of CO between homes; and differences between types of households, such as apartments or houses. The number of household occupants has been considered a theme affecting CO concentration levels, but it could also be considered a key aspect of population exposure measurement in epidemiology.

Many of the studies in this review did not include a model to understand CO household exposure, and only in one study was the model validated [39]. The most frequently used model was the mass balance model or a modification of this model. This model can be used to understand how CO disperses through a home, and it assumes that CO instantly becomes well mixed. It deviates from real-world measurements when near a source, and so modifications may be needed in these situations [42,43]. Studies that validate this model would help to confirm if this is an appropriate model in real-life situations and if the model’s assumptions still hold.

Whilst we aimed to identify studies that considered the health impacts of CO household exposure and the factors useful to characterise the health consequences of exposure, only one study did this [25]. This considered a variety of air pollutants and did not explore the relationship between only CO and health outcomes [25], so few conclusions can be drawn. The acute health impacts of CO are well known [3], and some evidence indicates that chronic low-level CO exposure may be associated with cardiovascular and respiratory morbidity and neurological impairment [12]. Further evidence is needed to understand the health effects of long-term exposure to the lower concentration levels seen in most homes. Further evidence is also needed on the exposure duration and number of people potentially exposed before an exposure approach to studying indoor CO health effects at lower CO concentration levels can be developed.

Our expectation is that in most homes in high-income countries, CO concentration levels are rarely high enough to observe adverse acute health effects if CO-producing appliances are not faulty and there is adequate ventilation. However, a small proportion of homes may have short-term high-level exposure or may have long-term low-level exposure at a level that has health consequences. Many of the studies in our review have small sample sizes, and detecting health impacts from long-term low-level exposure would require a very large sample of homes followed up over time. An improved understanding of how low levels of CO disperse in a home environment would also help to identify at-risk dwellings, and this could be provided by a dispersion study of CO emissions in a variety of homes.

### 4.1. Limitations

There are inherent limitations in the findings due to the design of this study, as it is not a systematic literature review. However, this rapid review followed as much as possible the steps of a systematic review and systematically used the PRISMA framework [21] to mitigate these shortcomings. The studies in our review are very heterogeneous, with some focused on the physical characteristics of CO and aimed at different scientific audiences than for epidemiological studies. Hence, narratively synthesising the evidence was challenging. Using artificial intelligence may have assisted with this and with identifying themes. In addition, the size and quality of the studies were variable, with some not clearly reporting study results. One study had to be excluded because our translation service was unable to translate it. Some studies did not specify the room in which CO was monitored or only measured it in one room, making a comparison of CO concentration levels between rooms difficult. Whilst we were focused on the household unit in our review, there was limited information in the different studies about the characteristics of the populations residing in households, such as age and SES, which may be important for characterising CO concentration levels. It should also be noted that our study focused on CO exposure within the household (at the group level) rather than on an individual person’s CO exposure, and it is recognised that individual exposure may differ. There was some limited evidence on CO models, but only one study used a model to understand CO exposure and validate it [39]. Due to the lack of evidence identified in this review about the health impacts of CO household exposure and the factors useful to characterise the health consequences of exposure, we are unable to draw meaningful conclusions about this in our review.

### 4.2. Recommendations

Some studies included in this review only measured CO over a short period of time [25,28]. Having studies that measure in the same households over longer periods and at different times of year will help us to better understand the impact of seasonal changes, how CO concentrations vary over a 24 h period due to the usage of household gas appliances, and how other factors affect CO concentration levels. CO alarms are usually triggered if levels are above 50 ppm for more than 60 min or if they are above 100 ppm for more than 10 min, so the alarms should have sounded in some of the homes in the studies in this review. In addition, having CO alarms that measure and record CO concentrations to much lower levels would aid epidemiological studies focused on the health impacts of chronic low-level CO exposure.

In the UK, CO safety messages are focused on installing home CO alarms and the maintenance of gas appliances, with only 46% of homes in England having a CO alarm in 2020 [45]. Consideration should also be given to raising awareness of the importance of ventilation when cooking, with studies demonstrating lower CO concentration levels in kitchens where an extractor fan was used [22,31], but also for other activities that may produce CO, such as an open fireplace, woodburning stove or burning charcoal [35]. In England, it is a legal requirement for landlords to have a CO alarm installed in every room used as living accommodation in rental properties that have a fixed combustion appliance, excluding a gas cooker [46]. Ensuring alarms are also in other rooms where CO may be produced, as well as in non-rental homes, would help to reduce risks further.

From this review, there is some evidence that the mass balance model is an appropriate model for understanding CO concentration levels in an epidemiological study considering household dwelling CO exposure. Studies that validate models for predicting CO concentration levels in homes would be an important contribution to understanding the applicability of the model in this setting.

When characterising CO concentration levels in household environments, factors impacting levels which should be considered are as follows: seasonality, with higher CO concentration levels in winter; the size of the room that measurement is taking place in, with higher levels seen in smaller rooms with a CO source; ventilation and outdoor CO concentration levels, with higher outdoor CO levels and greater ventilation from outdoors to indoors increasing CO concentration levels, and the use of extraction fans and AC lowering CO concentration levels. Cooking is an important factor resulting in raised CO levels in kitchens and in other rooms in a dwelling due to the dispersion of CO throughout homes [22,36]. Evidence for explanatory factors was thin, and there is a need for studies to consider the driving factors affecting CO concentration levels in homes, possibly through regression modelling or principal component analysis. In addition, studies and reviews that focus on an individual’s CO exposure within the home through personal exposure monitoring would be useful. Lastly, studies that consider the health impacts of long-term CO household exposure would be an important contribution to this topic and would inform public health actions, including both building design and individual behaviour, to reduce the health risks of CO household exposure.

## 5. Conclusions

Characterising CO exposure in household dwellings is key to understanding where CO exposures may result in adverse health outcomes, which can then be used to identify higher-risk households and inform public health actions. Wider use of CO alarms and ventilation would likely substantially reduce risks. We identified sixteen themes grouped into five main categories: dwelling characteristics, source characteristics, temporal variation, environmental characteristics, and socioeconomic status of occupants. The themes with the most evidence included seasonal variation (temporal variation), size of room and ventilation (dwelling characteristics), and cooking and outdoor CO levels (source characteristics). There was also some evidence for the benefit of using mass balance models to understand CO concentration levels in households, but there was little evidence reported about the health impacts of CO household exposure.

Hence, important areas for further research include examining the health impacts of CO household exposure and identifying factors promoting health consequences. Validating the use of the mass-balance model in household dwellings and the themes identified in this review, including those for which evidence is currently lacking, such as the socioeconomic status of household occupants, would be of value. These themes characterising CO exposure in household dwellings are important to aid the development of indoor exposure models and for understanding where and when CO exposures result in adverse health outcomes. These themes should be included in household CO monitoring studies, enabling the identification of higher-risk household dwellings and informing public health interventions to reduce health risks.

## Figures and Tables

**Figure 1 ijerph-22-00110-f001:**
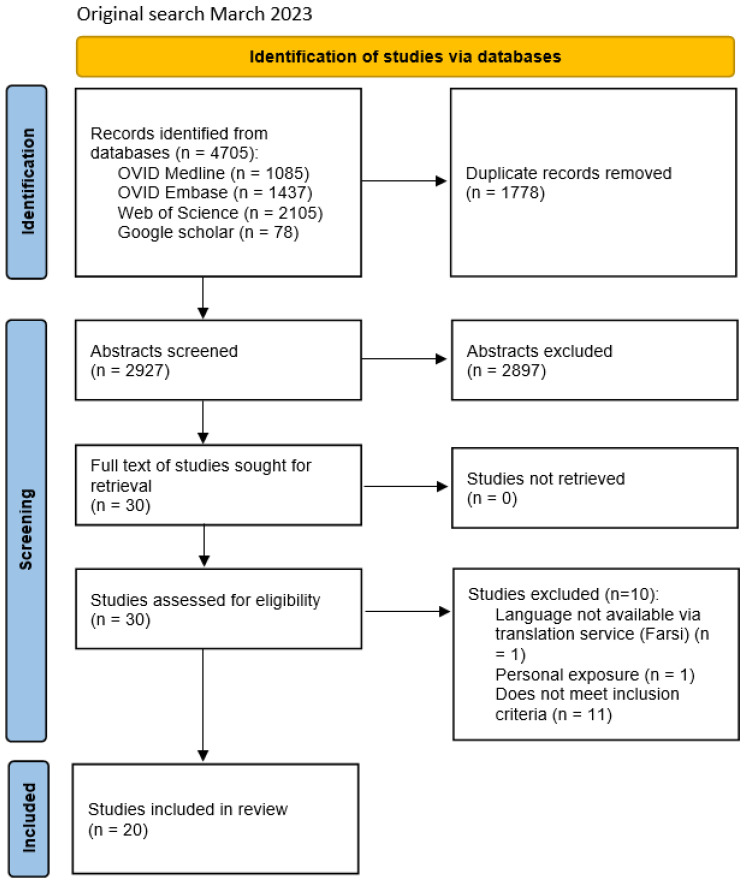
PRISMA flow diagram flowchart for the review.

**Figure 2 ijerph-22-00110-f002:**
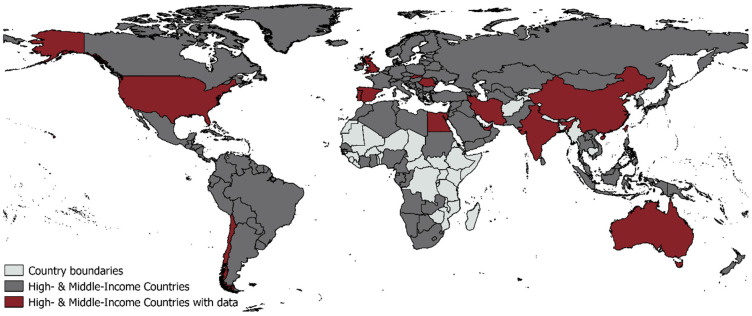
Countries that were the study locations of the papers included in the review. The figure is created by the authors and can be reproduced under the Open Government Licence (OGL). Not to scale.

**Table 1 ijerph-22-00110-t001:** Characteristics of papers.

	Number of Papers n (%)
Study designs	
Observational	17 (77)
Modelling	2 (9)
Modelling and experimental	2 (9)
Observational and modelling	1 (4.5)
Country	
USA	7 (32)
China	2(9)
Iran	2 (9)
Portugal	2 (9)
UK	2 (9)
Australia	1 (4.5)
Chile	1 (4.5)
Egypt	1 (4.5)
India	1(4.5)
Romania and Slovakia	1 (4.5)
Spain	1 (4.5)
UAE	1 (4.5)
Setting	
Urban or suburban	12 (55)
Mixed	6 (27)
Rural	3 (14)
Not specified	1 (4.5)
Housing type	
Various types included in study	9 (41)
Flat/Apartment/Condominium/Maisonette	5 (23)
House (detached, semi-detached, townhouse)	5 (23)
Not specified	3 (14)
Room(s) of study focus	
More than one room	7 (32)
Kitchen	4 (18)
Living room	4 (18)
Other	3 (14)
Not specified	3 (14)
Bedroom	1 (4.5)

**Table 2 ijerph-22-00110-t002:** Overview of individual studies, themes identified that characterise CO levels, and outcomes (grouped by study design and in alphabetical order of the lead author’s last name).

Study	Country	Design	Number of Homes	Number of Occupants	Overview	Duration of Measurement	Categories and Themes	Outcome: CO Concentration Levels *(Units as Reported in Study with Conversion to ppm *)*	Outcome: Exposure Models and Health Outcomes
Abdel-Salam [22]	Egypt	Observational	60	3.7 mean number of occupants	Investigates variations in levels of indoor pollutants, including CO, in two areas in Alexandria, Egypt, and the factors influencing concentration level variations.	2–3 occasions for 24 h each.	**Dwelling characteristics**: size of rooms, number of occupants, age of building, ventilation, distance to roads. **Source characteristics:** cooking, smoking, outdoor air quality. **Environmental characteristics:** temperature, relative humidity. **Socioeconomic status**	24 h mean concentration Study area 1: 1.1 ± 0.6 ppm outside, 1.0 ± 0.4 ppm in the living room and 1.7 ± 0.8 ppm in the kitchen. Study area 2: 1.2 ± 0.5 ppm outside, 0.7 ± 0.4 ppm in the living room and 0.6 ± 0.4 ppm in the kitchen.	
Canha et al. [23]	Portugal	Observational	1	1 occupant	Assessing indoor air quality, including CO, during sleep for different natural ventilation states	Approximately for 6 h.	**Dwelling characteristics:** ventilation.	CO concentration levels were always below the Portuguese limit value of 10 mg/m^3^ (8.6 ppm). The lowest mean CO level was 1.6 ± 0.3 mg/m^3^ (1.4 ± 0.3 ppm). Highest mean value was 3.8 ± 1.0 mg/m^3^ (3.3 ± 0.9 ppm).	
Castro et al. [24]	Spain	Observational	1	Uninhabited	To characterise the impact on indoor air quality, including CO, of the different stages of combustion from an open fireplace.	4–5 h for each experiment.	**Source characteristics:** fireplaces.		
De Almeida et al. [25]	Portugal	Observational	557	3 mean number of occupants	To assess indoor air quality, including CO, in homes across Portugal, focusing on the master bedroom and kitchen.	15 min in each home between December and July.	**Dwelling characteristics:** floor level, number of occupants, age of building. **Source characteristics:** fireplaces, smoking. **Environmental characteristics:** temperature, relative humidity.	Mean concentration of 1.2 mg/m^3^ (1.0 ppm) and median 1.1 mg/m^3^ (0.9 ppm) over 15 min. Only two measurements were above the Portuguese legal limits of 12.5 mg/m^3^ (10.7 ppm).	Related health outcomes
Francisco et al. [26]	USA	Observational	30	Not provided	Combustion products, including CO emitted from unvented fireplaces, under common usage and maintenance patterns.	For 3–4 days between December and March.	**Dwelling characteristics:** size of rooms, age of buildings. **Source characteristics:** fireplaces.	Maximum 1 h average recorded was 17.9 ppm, and the maximum 8 h average was 14.2 ppm at the same home. 1 h average CO in all homes was below the WHO guideline of 25 ppm, and 6 homes had 8 h averages greater than the EPA guideline of 9 ppm.	Models
Gilbey et al. [27]	Australia	Observational	111	60% of homes had ≥3 occupants	Household characteristics that influence indoor air quality, including CO concentration levels.	24 h.	**Temporal variations:** seasonal variations, daily variations. **Dwelling characteristics:** number of occupants, age of buildings, ventilation, distance to roads. **Source characteristics:** cooking, space heating and cooling homes.	24 h averaged concentrations of CO was 0.94 ± 0.1 ppm, below the WHO limits	
Majdan et al. [28]	Romania Slovakia	Observational	30	4 median number of occupants	To assess indoor air quality in Roma villages in Slovakia and Romania and identify implications for health.	For 6 min, repeated 1 h later. Each home was measured twice: once in winter and summer.	**Temporal variations:** seasonal variations. **Dwelling characteristics:** number of occupants. **Source characteristics:** fireplaces, smoking. **Environmental characteristics:** temperature.	Lower median CO concentrations were recorded in the 19 homes in Slovakia compared to those in Romania in both winter (2.4 (IQR 1.7–3.5) vs 4.9 (IQR 3.6–7.2) mg/m^3^) (2.1 (IQR 1.4–3.0) vs 4.2 (IQR 3.1–6.2) ppm) and summer (0.65 (IQR 0.4–1) vs 2.6 (IQR 2.1–3) mg/m^3^) (0.56 (IQR 0.3–0.9) vs 2.2 (IQR 1.8–2.6) ppm).	
Martinez et al. [29]	Chile	Observational	96	Not provided	To assess exposure to CO and risk factors for people living in apartment buildings.	12 h between September and December.	**Dwelling characteristics:** ventilation. **Source characteristics:** cooking, smoking, outdoor air quality.	Median indoor CO concentration was monitored at 1 (IQR 0–1) ppm.	
Mfarrej et al. [30]	UAE	Observational	12	Not provided	Assessing residential indoor air quality in UAE and how to improve indoor air quality.	Every hour for 8 h.		Recorded values ranging from 0 to 5.61 ppm measured at the side door, kitchen, and bathroom and measured every hour for 8 h.	
Mullen et al. [31]	USA	Observational	316	49% of homes had ≥3 occupants	Assessing how natural gas appliances impact air quality in homes.	Monitored every minute over 6 days in winter.	**Dwelling characteristics:** number of occupants, ventilation. **Source characteristics:** cooking. **Socioeconomic status**	The mean and geometric mean of the highest 1 h CO concentration in each home were 6.4 ppm and 3.8 ppm.	
Naghizadeh et al. [32]	Iran	Observational	25	Not provided	To assess indoor and outdoor CO concentration levels in Sarayan, Iran.	Monitored in the kitchens for 10 days per month between January and March and the hours 11.00 and 15.00.	**Source characteristics:** outdoor air quality.	Mean CO levels were 0.84 ± 3.21 ppm.	
Nandan et al. [33]	India	Observational	5	4.8 mean number of occupants	To investigate indoor air quality in Dehradun, India.	Unclear.	**Dwelling characteristics:** number of occupants, ventilation. **Source characteristics:** cooking.	Mean CO levels across all homes were 8.90 ppm.	
Pickett et al. [34]	USA	Observational	10	Not provided	To assess indoor air quality in homes with infants and to consider variation of pollutants within homes, and potential sources and associations.	Measured between 4 and 7 days between June and August.	**Environmental characteristics:** temperature, relative humidity.	Hourly mean values were 0.85 ppm, median 0.79 ppm with a minimum value of 0.25 ppm and a maximum value of 4.69 ppm, below WHO guidelines.	
Qin et al. [35]	China	Observational	35	Not provided	To improve understanding of the characteristics of indoor air pollution caused by coal burning.	Monitored for between 2 and 2.5 h.	**Dwelling characteristics:** ventilation. **Source characteristics:** fireplaces, outdoor air quality.	Approximately 91% of the households had hourly mean CO concentrations (2.7–377.7 ppm) above the national standard of 8.75 ppm during the heating period. Mean concentrations across the fireplace combustion process were 85.4 ± 80.6 ppm. During the non-heating period, the mean concentrations were 2.8 ± 1.2 ppm.	
Shen et al. [36]	China	Observational	1	2 occupants	Determine major sources of CO and how concentration levels vary horizontally, vertically and temporally.	Every 10–20 min for between 5 and 14 days, depending on the location of monitors.	**Dwelling characteristics:** size of rooms. **Source characteristics:** cooking, outdoor air quality.	Indoor concentrations of CO were 1.6 ± 0.99 ppm. In the living room of the home, all hourly mean concentrations were lower than the national standard of 8.75 ppm, but in the kitchen, peaks of 20–30 ppm were briefly seen, though hourly mean concentrations were less than the standard.	
Stamp et al. [37]	UK	Observational	5	4.8 mean number of occupants	To assess the presence of pollutants in urban apartments and the role of ventilation and seasonal variation in indoor air quality.	Measured twice over heating and non-heating seasons for 7 days.	**Temporal variations:** seasonal variations. **Source characteristics:** outdoor air quality.	The median CO concentration was 168 µg/m^3^ (0.14 ppm) and the mean was 183 µg/m^3^ (0.16 ppm), with the minimum value recorded at 129 µg/m^3^ (0.11 ppm) and the maximum value at 343 µg/m^3^ (0.29 ppm).	
Tan et al. [38]	UK	Observational	3	Not provided	To investigate concentration levels and emission rates of indoor air pollutants, including CO, from various heating and cooking systems.	7 days (not stated; information inferred from a figure).	**Temporal variations:** daily variations. **Dwelling characteristics:** ventilation. **Source characteristics:** cooking, outdoor air quality.	The average CO concentration in each home’s kitchen was low at between 1.0 and 2.0 ppm, although spikes in CO concentration levels were seen in the two flats during cooking periods.	Models
Ardeh et al. [39]	Iran	Observational Modelling			Estimating the effect of natural gas burners on indoor air quality in residential buildings.		**Temporal variations:** seasonal variations, daily variations **Dwelling characteristics:** ventilation.		Models
Logue et al. [40]	USA	Modelling			To quantify pollutant concentration levels and occupant exposures from natural gas cooking burners in homes.				Models
Whitmyre et al. [41]	USA	Modelling			To estimate the impact of vent-free gas heating on indoor concentration levels of CO through probabilistic analysis.				Models
Acevedo-Bolton et al. [42]	USA	Modelling Experimental	2	Not provided	Investigating the proximity effect of CO source in a naturally ventilated home.	For between 5 and 12 h, depending on the location.	**Dwelling characteristics:** ventilation.	Maximum mean background adjusted CO concentration of 48 ppm. at a radial distance of 0.25 m from the source.	Models
Cheng et al. [43]	USA	Modelling Experimental		Not provided	Using the eddy diffusion model, investigate the isotropic turbulent diffusion coefficient (K) in naturally ventilated residences of a continuous CO source and how well the model predicts CO concentration levels at different distances from a source.	For between 5 and 12 h, depending on location (as per information from [42]).	**Dwelling characteristics:** ventilation.		Models

* mg/m^3^ converted to ppm using the conversion 1 ppm = 1.165 mg/m^3^ [14].

**Table 3 ijerph-22-00110-t003:** WHO CO guidance levels [11].

Averaging Time	Concentration (mg/m^3^)	Concentration (ppm) *
15 min	100	86
1 h	35	30
8 h	10	8.6
24 h	7	6

Adapted from: WHO guidelines for indoor air quality: selected pollutants, 2010. * mg/m^3^ converted to ppm using the conversion 1 ppm = 1.165 mg/m^3^ [14].

**Table 4 ijerph-22-00110-t004:** Identified categories and themes to characterise CO exposure.

Category	Theme
Temporal variations	Seasonal variation Daily variations
Environmental characteristics	Temperature Relative humidity
Dwelling characteristics	Floor level Size of rooms Number of occupants Age of building Ventilation (air exchange rate) Distance to roads
Source characteristics	Cooking Space heating and cooling homes Fireplaces Smoking Outdoor air quality affecting indoor CO levels
Socioeconomic status of occupants	Socioeconomic status of occupants

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
