# Peer review of "Characterising Carbon Monoxide Household Exposure and Health Impacts in High- and Middle-Income Countries—A Rapid Literature Review, 2010–2024"

_ijerph, 2025, doi:10.3390/ijerph22010110_

Round 1

Reviewer 1 Report

Comments and Suggestions for Authors

A very good review, especially about ventilation, cooking and fireplaces.  

I have four trivial corrections/ comments:

Line 41: you should state fossil fuels and biomass in combustion sources.

Table 2: each column is equal width. If you compressed columns 2-5 then there would be more width for columns 6,8,9 for easier reading.

Line 389: "gas cookers or when natural gas was used": what is the difference between these two sources in HMICs?

Line 405: for clarity, change The houses to These houses

I have four issues that I believe need correcting or commenting:

You state that 1ppm = 1.165mg/m3.  The EC uses 1.1642mg/m3, based on 20C, 2013hPa. The WHO uses 1.15 mg/m3, based on 25C, 1013 hPa. You should use 1.1642 (1.165 is fine), but attribute it to EC, not WHO.

Line 325 and later: you use AER, which is unusual and not well defined. I am used to ACH and do not recollect others using AER as a common ventilation term. I recommend you change the text to ACH (some people prefer ACR).

The introduction lists CO mortality rates but the review is about ambient, "safe" CO concentration levels. The reader starts the review, expecting a review of unsafe CO concentrations, but the review considers very low CO levels.  I do not have a solution to this contradiction, perhaps emphasise more strongly in the introduction that this review is not about CO leaks from faulty gas boilers, which are the main source of CO deaths.

Your recommendations are very good, and there is one more to add:  if looking at the long term health implications requires large scale epidemiological studies, then why not convince the domestic CO alarm suppliers to improve their products so they can resolve down to 0.2 or 0.5 ppm and log and report the results? There are millions of CO alarms and why can they not be used to record a large cohort if the suppliers improved their alarms?

Author Response

Response to reviewer 1

Thank you for reviewing our manuscript and for your comments. Please find below our detailed responses to your comments and the revisions in tracked changes in the resubmitted manuscript.

Comments and Suggestions for Authors

A very good review, especially about ventilation, cooking and fireplaces. 

I have four trivial corrections/ comments:

Comment 1

Line 41: you should state fossil fuels and biomass in combustion sources.

Thank you, this has been updated to read as (line 42) “…by-product from incomplete combustion of fossil fuels or biomass sources.”.

Comment 2

Table 2: each column is equal width. If you compressed columns 2-5 then there would be more width for columns 6,8,9 for easier reading.

We have compressed columns 2-5, to give more space for later columns.

Comment 3

Line 389: "gas cookers or when natural gas was used": what is the difference between these two sources in HMICs?

Some of the studies included in the review are from middle-income countries, where natural gas is more commonly used. We have updated the text to read as (line 409) “… particularly when natural gas was used as a fuel source.

Comment 4

Line 405: for clarity, change The houses to These houses

This sentence has been updated to (line 426) “These homes…”.

I have four issues that I believe need correcting or commenting:

Comment 5

You state that 1ppm = 1.165mg/m3.  The EC uses 1.1642mg/m3, based on 20C, 2013hPa. The WHO uses 1.15 mg/m3, based on 25C, 1013 hPa. You should use 1.1642 (1.165 is fine), but attribute it to EC, not WHO.

We have used the 2021 WHO global air quality guidelines for carbon monoxide which use the conversion factors as per the manuscript text (page 130 of the pdf WHO global air quality guidelines: particulate matter (‎PM2.5 and PM10)‎, ozone, nitrogen dioxide, sulfur dioxide and carbon monoxide) so have kept the conversions in the text the same as originally stated in the manuscript (footnote table 2 and table 3).

Comment 6

Line 325 and later: you use AER, which is unusual and not well defined. I am used to ACH and do not recollect others using AER as a common ventilation term. I recommend you change the text to ACH (some people prefer ACR).

We have updated the text to ACH (line 343 onwards) and have added a brief explanation of the different terms to the first paragraph in the ventilation section because some of the cited literature uses AER:

Ventilation affects the inflow of pollutants from outside and the removal of pollutants from indoors (Vardoulakis et al., 2020). It is sometimes referred to as the air exchange rate (AER) in the literature, the airflow rate per volume of the indoor environment, or more commonly the air change per hour (ACH), which we use here.

Comment 7

The introduction lists CO mortality rates but the review is about ambient, "safe" CO concentration levels. The reader starts the review, expecting a review of unsafe CO concentrations, but the review considers very low CO levels.  I do not have a solution to this contradiction, perhaps emphasise more strongly in the introduction that this review is not about CO leaks from faulty gas boilers, which are the main source of CO deaths.

The introduction has been updated to strengthen this point and now includes the following text (line 73):

In the UK, ambient household levels of CO are reported to be low and below the World Health Organization’s (WHO) recommended health based guidelines (UK Health Security Agency, 2022; World Health Organization, 2010). However, there is still a risk of chronic low-level exposure in the home, which may be associated with respiratory and cardiovascular morbidity (UK Health Security Agency, 2022).

And (lines 96-98):

Therefore, identifying themes that characterise CO exposure in household dwellings is key to understanding where CO exposures may result in adverse health outcomes, including for chronic low-level exposure.

Comment 8

Your recommendations are very good, and there is one more to add:  if looking at the long term health implications requires large scale epidemiological studies, then why not convince the domestic CO alarm suppliers to improve their products so they can resolve down to 0.2 or 0.5 ppm and log and report the results? There are millions of CO alarms and why can they not be used to record a large cohort if the suppliers improved their alarms?

Thank you for this recommendation. We have updated the recommendations section of the manuscript to include (lines 670 – 672):

In addition, having CO alarms that measure and record CO concentrations to much lower levels, would aid epidemiological studies focused on the health impacts of chronic low-level CO exposure.

Reviewer 2 Report

Comments and Suggestions for Authors

The introduction should clearly articulate the dangers of carbon monoxide (CO) and the necessity for this research. Incorporating global or country-specific CO exposure statistics can capture the reader's attention. Additionally, when outlining the study's objectives, it is advisable to present them in bullet points for quick comprehension.

1.                 In the results section, summarizing each topic can be made more accessible by adding subheadings for each category. This will enhance readability and facilitate the location of specific information. Furthermore, when presenting measurements of CO concentration and WHO standards, improving the aesthetics and clarity of the tables will aid in conveying the data effectively.

2.                 Clarifying the timeframe of selected literature (2010-2024) and the criterion for inclusion is crucial. This will underscore the relevance and importance of the chosen studies.

3.                     Statistical analysis should also be included when discussing CO concentration levels. Providing specific results—such as means, standard deviations, and confidence intervals—will deepen the reader's understanding beyond mere descriptive data.

4.                 In the health effects discussion, it would be beneficial to expand on the potential impacts of long-term low-level CO exposure, supported by relevant literature. Maintaining consistency in terminology throughout the manuscript, for instance, using “CO” or “carbon monoxide” uniformly, is essential for clarity.

5.                  Incorporating charts or graphs where appropriate will help visualize data trends, such as seasonal variations in CO concentrations. Lastly, the conclusion should clearly outline priority areas for future research, particularly concerning the health effects of CO exposure and recommendations for enhanced monitoring methods. Specific public health recommendations, like promoting the use of CO alarms and improving ventilation, can further enhance the practical relevance of the study.

Author Response

Response to reviewer 2

Thank you for reviewing our manuscript and for your comments. Please find below our detailed responses to your comments and the revisions in tracked changes in the resubmitted manuscript.

The introduction should clearly articulate the dangers of carbon monoxide (CO) and the necessity for this research. Incorporating global or country-specific CO exposure statistics can capture the reader's attention. Additionally, when outlining the study's objectives, it is advisable to present them in bullet points for quick comprehension.

Thank you, we have updated the introduction so that the objectives are presented as bullet points (lines 105-115). Information on the dangers of carbon monoxide and the necessity of research are included in the introduction from line 48 onwards, and information on global and country-specific carbon monoxide exposure statistics are included in the introduction from line 63-69.

  1. In the results section, summarizing each topic can be made more accessible by adding subheadings for each category. This will enhance readability and facilitate the location of specific information. Furthermore, when presenting measurements of CO concentration and WHO standards, improving the aesthetics and clarity of the tables will aid in conveying the data effectively.

In the results section, we have presented the five categories and 16 themes with subheadings, which should be clearer in this reformatted version of the manuscript. We have also reduced the widths of columns 1 to 5 in table 2, so that it is easier to read the later columns which have more text.

  1. Clarifying the timeframe of selected literature (2010-2024) and the criterion for inclusion is crucial. This will underscore the relevance and importance of the chosen studies.

Thank you, we have explained the timeframe chosen and clarified the inclusion criteria, so that the text now reads as below:

(line 126) These dates were chosen to ensure that selected literature was most representative of households and heating and cooking habits of current times.

And:

(line 132) We included studies in this review that:

  • Contained modelling or monitoring of CO concentration levels, sources, emissions, or related health effects;
  • Attempted to build, use, validate or evaluate a model for such exposure or for its use for human health exposure assessment and Health Impact Assessment;
  • Were set in the indoor home environment;
  • Were set in a high or middle-income country (HMIC).

We had no language restrictions. Our preference was for studies that have measured or modelled CO in multiple homes. However, studies in single homes were included if they otherwise met the inclusion criteria.

  1. Statistical analysis should also be included when discussing CO concentration levels. Providing specific results—such as means, standard deviations, and confidence intervals—will deepen the reader's understanding beyond mere descriptive data.

The purpose of this study is to present and review the literature on this topic through narrative synthesis. As highlighted in the synthesis of data and limitations sections, the studies are very heterogenous and it is not possible to conduct statistical analysis beyond what the studies have presented themselves, which we have included where available in table 2.

  1. In the health effects discussion, it would be beneficial to expand on the potential impacts of long-term low-level CO exposure, supported by relevant literature. Maintaining consistency in terminology throughout the manuscript, for instance, using “CO” or “carbon monoxide” uniformly, is essential for clarity.

Thank you, we have now included the potential impacts of chronic low-level CO exposure in the introduction and in the discussion section.

Introduction paragraph 4 (line 73):

In the UK, ambient household levels of CO are reported to be low and below the World Health Organization’s (WHO) recommended health based guidelines (UK Health Security Agency, 2022; World Health Organization, 2010). However, there is still a risk of chronic low-level exposure in the home, which may be associated with respiratory and cardiovascular morbidity (UK Health Security Agency, 2022). Hence, CO poisoning is likely to have a bigger public health impact than the official figures show.

Discussion paragraph 5 (line 622):

The acute health impacts of CO are well known (Bleecker, 2015), and some evidence indicates that chronic low-level CO exposure may be associated with cardiovascular and respiratory morbidity, and neurological impairment (UK Health Security Agency, 2022). Further evidence is needed to understand the health effects from long-term exposure to the lower concentration levels seen in most homes.

We have defined the abbreviation of carbon monoxide as CO in the introduction (line 42) and have used CO thereafter. Where a title of a report contains the words “carbon monoxide”, we have not changed this to CO.

  1. Incorporating charts or graphs where appropriate will help visualize data trends, such as seasonal variations in CO concentrations. Lastly, the conclusion should clearly outline priority areas for future research, particularly concerning the health effects of CO exposure and recommendations for enhanced monitoring methods. Specific public health recommendations, like promoting the use of CO alarms and improving ventilation, can further enhance the practical relevance of the study.

We are not including graphs or charts of the data from papers as we do not have copyright and the purpose of the manuscript was to narratively synthesize information from the literature, not to replicate it. Much of the information narratively synthesised from the included studies are not appropriate to display graphically as the studies measure over different time periods or different units/contexts.

We have updated the recommendations paragraph 1 (line 670) to include alarms and monitoring at low-levels of CO:

In addition, having CO alarms that measure and record CO concentrations to much lower levels, would aid the development of epidemiological studies focused on the health impacts of chronic low-level CO exposure.

We have included recommendations for promoting the use of CO alarms and improving ventilation in the conclusion and in paragraph 2 (line 673) of the recommendations section.

Round 2

Reviewer 2 Report

Comments and Suggestions for Authors

Sarah V. Williams and coworkers described characterizing carbon monoxide household exposure and health impacts in high- and middle-income countries. The discussion about structural configuration and literature review were detailed. Overall, it could be suitable for Int. J. Environ. Res. Public Health in current form (after good revision).